# Edible Chitosan Films and Their Nanosized Counterparts Exhibit Antimicrobial Activity and Enhanced Mechanical and Barrier Properties

**DOI:** 10.3390/molecules24010127

**Published:** 2018-12-31

**Authors:** Laidson P. Gomes, Hiléia K. S. Souza, José M. Campiña, Cristina T. Andrade, António F. Silva, Maria P. Gonçalves, Vania M. Flosi Paschoalin

**Affiliations:** 1REQUIMTE/LAQV, Departamento de Engenharia Química, Faculdade de Engenharia, Universidade do Porto, Rua Dr. Roberto Frias, 4200-465 Porto, Portugal; laidsonpaes@gmail.com (L.P.G.); hsouza@fe.up.pt (H.K.S.S.); pilarg@fe.up.pt (M.P.G.); 2Instituto de Química, Universidade Federal do Rio de Janeiro, Av. Athos da Silveira Ramos 149, Cidade Universitária, 21941-909 Rio de Janeiro, Brazil; 3Departamento de Química e Bioquímica, Centro de Investigação em Química da Universidade do Porto (CIQ-UP), Faculdade de Ciências, Rua do Campo Alegre, 687, 4169-007 Porto, Portugal; jpina@fc.up.pt (J.M.C.); afssilva@fc.up.pt (A.F.S.); 4Instituto de Macromoléculas Professora Eloisa Mano, Universidade Federal do Rio de Janeiro, Av. Horácio Macedo, 2030-Cidade Universitária, 21941-598 Rio de Janeiro, Brazil; ctandrade@ima.ufrj.br

**Keywords:** fishery byproduct, sonication, nanoparticles, high performance biobased films, scanning electron microscopy, green chemistry, eco-friendly and bioplastic

## Abstract

Chitosan and chitosan-nanoparticles were combined to prepare biobased and unplasticized film blends displaying antimicrobial activity. Nanosized chitosans obtained by sonication for 5, 15, or 30 min were combined with chitosan at 3:7, 1:1, and 7:3 ratios, in order to adjust blend film mechanical properties and permeability. The incorporation of nanosized chitosans led to improvements in the interfacial interaction with chitosan microfibers, positively affecting film mechanical strength and stiffness, evidenced by scanning electron microscopy. Nanosized or blend chitosan film sensitivity to moisture was significantly decreased with the drop in biocomposite molecular masses, evidenced by increased water solubility and decreased water vapor permeability. Nanosized and chitosan interactions gave rise to light biobased films presenting discrete opacity and color changes, since red-green and yellow-blue colorations were affected. All chitosan blend films exhibited antimicrobial activity against both Gram-positive and Gram-negative bacteria. The performance of green unplasticized chitosan blend films displaying diverse morphologies has, thus, been proven as a potential step towards the design of nontoxic food packaging biobased films, protecting against spoilage microorganisms, while also minimizing environmental impacts.

## 1. Introduction

A worldwide interest in replacing oil-based synthetic plastic packaging by biodegradable, nontoxic and edible materials is noted. The development of new packaging products can benefit several industrial activities, particularly food production, distribution, commercialization, and preservation [1]. Chitosan, a polysaccharide derived from chitin, is a promising biopolymer, due to its biocompatibility with animal cells and tissues, biodegradability and ability to form films. Chitin is an abundant polysaccharide in nature and can be obtained as a byproduct of the seafood industry or associated with shrimp aquaculture production. In addition to environmental benefits [2], chitosan is considered a potential active packaging material regarding the replacement of petroleum-based films, since it exhibits diverse bioactivities [3]. Physicochemical properties resulting from variations in chitosan molecular weight and degree of deacetylation can confer special functionalities to this biopolymer, which can be very useful in food packaging [4,5]. In addition, the chitosan polycationic character allows for complexation with natural antibiotics or antioxidants [6], reinforcing its applicability in food preservation [7].

The fabrication of transparent chitosan biobased films for food packaging and other applications has been widely studied [8]. Unfortunately, pure chitosan biobased films suffer from poorer mechanical properties and exacerbated moisture sensitivity compared to their oil-derived counterparts [9]. The use of certain plasticizers (for example, ethylene glycol) can bypass some of these limitations [10]. However, plasticizers can be transferred to food, thus posing a health threat. The addition of nanoparticles to chitosan biobased films may be a successful approach to improve the physicochemical and mechanical properties of chitosan films. In fact, the use of nanoparticles, in particular chitosan nanoparticles, is often reported as improving the mechanical and barrier properties of other polysaccharide films, composed of pectin [11] or starch [12].

However, films prepared solely from chitosan nanoparticles obtained from chitosan with different molecular masses and degree of deacetylation (*DD* = 90 and 95%) showed a significant decrease in permeability. However mechanical properties were adversely affected by polysaccharide fragmentation due to ultrasonication [13].

In a previous study, nanosized chitosans were prepared by high-M chitosan sonication and mixtures were prepared by nonsonicated and ultrasonic fragments demonstrating that it is possible to adjust viscoelasticity and morphological characteristics of the mixtures at intermediate levels among those presented by their individual components. The high-M-chitosan mixtures its ultrasonic fragmentation derivatives point towards the development of a novel generation of additive-free sustainable but functional biobased films. However, the challenge remains to produce films with desirable mechanical strength, despite a putative weakness brought by the ultrasonic fragmentation of the polysaccharide molecule [14].

The aim of the present study was to assess if aqueous blends composed of chitosan and nanosized chitosan, at different ratios, could be useful in the development of edible food package films with fine-tuned properties, without the inclusion of plasticizers that could introduce toxic or non-biodegradable components to the chitosan films. The physical, functional, and barrier properties of the blend films concerning the amount of each blend component were investigated. The antimicrobial activity of each blend film was tested against food spoilage microorganisms.

## 2. Results and Discussion

### 2.1. Film Thickness and Mechanical Properties

Using the knife-coating technique, homogeneous, transparent, and flexible films prepared without the use of plasticizers were obtained from nonsonicated chitosan (NS), sonicated chitosan (S), and colloidal blends (NS/S). In general, the thickness of the NS films was smaller than those prepared with S and NS/S blend films (Table 1). Mechanical properties, including tensile strength, Young’s modulus, and elongation at break, of the chitosan-based films were evaluated and are displayed in Figure 1.

In general, NS/S films exhibited enhanced mechanical resistance when compared to their sonicated precursors (S) (Figure 1), the stress curve for the sample NS/S_5min_, 7:3 is showed at Appendix A. Variable tensile strength was observed, depending on the blend composition, where improved tensile strengths were observed for the NS/S_5min_, at a 3:7 ratio blend, the variability between the values can be observed in Appendix A. NS/S_15min_ films showed increased resistance at 7:3 and 1:1 ratios when compared to NS. However, decreased tensile strength was observed for the 3:7 blend films. Discrete variations in maximum stress were observed for the NS/S_30min_ film formulations.

The ability of the films to stretch (elongation at break) also varied between the distinct blends (Figure 1B). In general, the elongation at break value for the NS film was lower than for the NS/S_5min_ and NS/S_15min_ blend films. NS/S_30min_ at 7:3 and 1:1 blend ratios displayed a similar elongation at break value to NS films. In other words, the elongation at break measures was enhanced for NS/S_5min_ and NS/S_15min_ at different blend ratios, while a decrease for NS/S_30min_ was observed.

Regarding the elastic modulus (Young’s modulus), three groups displaying distinct properties were observed (Figure 1C): (i) films prepared with the NS/S_5min_ blend presented lower Young’s modulus values than NS films and similar values to the S_5min_ precursor; (ii) the opposite was observed for films prepared with NS/S_15min_; and (iii) NS/S_30min_ films presented Young’s modulus values closer to that of the NS film and three-fold higher than for the S_30min_ film. However, increases S content in the film formulations led to small differences in the Young’s modulus within each group.

Since no plasticizers were added to control the film formation process, the distinct behaviors observed for the blend films (NS/S) may be ascribed to their composition. In fact, as demonstrated previously [13,14], chitosan-like microfibers were filled with nanoparticles (NPs), measuring 10–30 nm (smallest) and 150–300 nm (medium-size) where fiber width decreases led to increasing NPs, mainly medium-sized NPs [14]), observed during chitosan fragmentation by ultrasonication. Thus, the balance of the mechanical properties is probably due to the presence of NPs acting as reinforcing agents in the blend films, at different ratios. In fact, the films presenting the highest tensile strength values were NS/S_15min_ at blend ratios 7:3 and 1:1 (Figure 1). According to Gomes et al., 2016 [14], an increase in the number of medium-sized NPs relative to microfibers occurs when chitosan (NS) is sonicated for 15 min (S_15min_). Thus, mixing both formulations, at the 7:3 and 1:1 ratios, seems to promote a structural equilibrium in these films, leading to higher tensile strength (approximately 1.5-fold higher than for the S_15min_ precursor). In addition, these blend films displayed higher elongation at break values (two/three-fold higher than the NS film), whereas Young’s moduli were higher than for the NS and S precursors (Figure 1). On the other hand, the increased amount of NPs in the NS/S_15min_ blend at a 3:7 ratio led to mechanical property destabilization.

The increases in elongation at break values observed for the blends when compared to the NS could be indicative of decreased blend film brittleness. This is expected, since it has been well-documented that increases in elongation at break with the addition of a rigid filler are related to differences in rigidity between the matrix and the fillers [14]. The addition of chitosan nanoparticles may have affected the discontinuity of the polymer network, thus increasing NS/S composite film stretching capability and mobility.

The use of distinct nanoparticle proportions (NS/S) in the film composition led to improvements in their physicochemical characteristics, displaying potential to substitute commonly used petroleum-based plasticizers.

As no plasticizers were added, which could be transferred to packaged food, these improvements in NS/S film properties should broaden their use for the fabrication of biobased films for regular use in food, such as biofuel plastic substitutes, due to the GRAS status of chitosan.

### 2.2. Water Vapor Permeability and Moisture Sorption Isotherms

The water vapor permeability measures obtained for the blend films ranged from 3.9–13.2 × 10^−11^g∙m^−1^∙s^−1^∙Pa^−1^ (Table 1), lower than that observed for the NS films. NS/S_15min_ films at the 7:3 ratio displayed WVP (13.2 × 10^−11^ g∙m^−1^∙s^−1^∙Pa^−1^) similar to the nonsonicated precursor films (NS) and the NS/S_30min_ blends, the lowest WVP measures 3.9–8.7 × 10^−11^ g∙m^−1^∙s^−1^∙Pa^−1^, reaching the smaller value at the ratio blend of 3:7. Moreover, in the latter case, water vapor permeability decreased with increasing S_30min_ amounts in the blend film. The lowest value was observed for the 3:7 ratio blend (3.9 × 10^−11^ g∙m^−1^∙s^−1^∙Pa^−1^). Therefore, films incorporating the NS/S blend displayed improvements in water vapor permeability properties when compared to the NS film, with the best results observed for the NS/S_30min_ film. Indeed, as displayed in Table 1, increasing S_30min_ contents in the blend resulted in decreased water vapor permeability, which may be related to blend film morphology. These results corroborate those obtained for particle size distribution reported in our previous study [14]. According to the authors, sonication degrades chitosan microfibers at an apparent constant rate, mainly producing medium-sized NPs, at a similar rate, degradation seems be completed at sonication time <25 min, since beyond this time, the increment in chitosan microfiber content is almost negligible. Thus, the present results may indicate that the nanosized chitosan samples sonicated for 30 min (S_30min_) should reach the minimum size particle, which is able to fill the voids shaped by the NS chitosan matrix, where permeation of water molecules is hampered. Because of this, films incorporating both NS and S chitosan and NS/S films display improved moisture resistance, restricting and limiting water molecule incorporation. These results are in line with the morphology of the chitosan-blend films (Figure 3).

Relative humidity effects on film characteristics can be assessed by the analysis of the moisture sorption isotherms (Figure 2). The curves were sigmoidal, typical of hydrophilic materials. The calculated equilibrium moisture content, X_e_, increased above a_w_ = 0.6 for precursor NS and S films and above a_w_ = 0.3 for NS/S blends. Increased sonication times led to decreases in the water fraction adsorbed by S films, under high humidity conditions (Figure 2). In the case of the blend films, important changes were observed. For NS/S_5min_ and NS/S_15min_ films, water sorption increased for all blend ratios and was higher than for the respective precursors at any a_w_ range (Figure 2A,B). The opposite was observed for the NS/S_30min_ films (Figure 2C). The NS/S_30min_ films displayed the lowest moisture content at a given a_w_. The GAB model was used to fit the experimental water adsorption data [15]. The fitting parameters are displayed in Table 1. Overall, the model seems to adequately fit the experimental data, as suggested by the correlation coefficients (R^2^ > 0.98) and k values (0 < k < 1).

The amount of water retained at the primary sorption sites (monolayer) of the films was quantitatively described, on a dry basis, by X_0_. According to the qualitative curve analysis, an increase in the predicted X_0_ values was observed for the NS/S_15min_ and NS/S_30min_ blend films when compared to their nanosized precursors (Table 1). This may be indicative of the formation of enhanced open film morphologies in the blend films, increasing the number of active sites. The set of NS/S_5min_ blend films does not show differences in the predicted X_0_ values when compared to the nanosized precursor. However, a decrease in the amount of water adsorbed at the monolayer is noted in comparison to the NS films.

The strength of the bonds established between water molecules and hydrophilic sites at the film monolayer can be quantified through the Guggenheim constant C, where, the higher the C value, the stronger the bonds [16]. As observed in Table 1, C values decreased with increasing amounts of nanosized chitosan S_5min_ or S_15min_ incorporated to the NS/S blend films. This suggests that water adsorption by these films depends on the blend ratio and evolves from a monolayer of molecules more strongly bound to the material (higher C value) to a situation where the adsorption of one monolayer is not favored concerning further multilayer formation (lower C value). In contrast, NS/S_30min_ exhibited the lowest C values, since water molecules are less strongly bound to the material when it is organized as a monolayer.

### 2.3. Chitosan-Blend Film Morphology

Different arrangements are observed in the scanning electron microscope micrographs of the fractured surface of blend films prepared with NS/S (*t_S_* = 5, 15 or 30 min) at a 1:1 blend ratio (Figure 3). The incorporation of nanosized particles into the chitosan-fiber film matrix resulted in films with a high number of small discrete nanoparticles within the microfibers, more clearly observed in Figure 3A,B.

As the number/size of nanoparticles depends on sonication time, the image of the NS/S films obtained at a 1:1 blend ratio with S_15min_ or S_30min_ (Figure 3B,C) indicates more homogeneous and smooth cross-sections. The addition of S_15min_ or S_30min_ nanosized chitosans led to improvements in interfacial interactions with the chitosan microfibers.

These results are in line with the film mechanical properties and water vapor permeability/sorption features. As discussed previously, a significant improvement in mechanical properties (tensile strength, elongation at break, and Young’s modulus) was observed for the NS/S_15min_ films at a 1:1 blend ratio when compared to their precursors. Nevertheless, the blend prepared with NS/S_30min_ at the same ratio also resulted in films with improved mechanical properties when compared to their nanosized precursor, although they displayed the lowest water vapor permeability values, since less active sites are available to bind to water molecules.

### 2.4. Color Properties

To evaluate the color and opacity of the films developed herein, color differences (∆E) were assessed taking into account the following parameters: lightness (L), red-green (a), and yellow-blue (b) components (Table 2).

All chitosan biobased films showed lightness with L coordinate ≥ 91 (Table 2). However, significant differences were observed when S_15min_ or S_30min_ chitosan were added to the NS chitosan in blend films, since they affected a* (red-green) and b* (yellow-blue) values. All blend films showed changes in a*, when compared to the NS precursor, except for the blends NS/S_5min_, NS/S_10min_ and NS/S_15min_ at a 3:7 ratio blend. In general the introduction of nanosized chitosan increased the yellow-blue coloration of films. High b* values (higher yellowness) were observed for NS/S_15min_, at 7:3 and 1:1 ratios, which presented values of 91.30 ± 0.8 and 92.65 ± 0.8, respectively. These films were also the darkest, presenting the lowest L values, as observed in Table 2. The addition of nanosized chitosan provoked a similar effect described for plasticizers in chitosan biobased films, where the addition of glycerol and Tween 20 led to lightness and yellowness decreases in chitosan films [17,18].

Sonication time led to significant effects on total color differences when comparing NS and S films. Sonication for 5 and 15 min decreased or maintained ∆E, however after 30 min, the S_30min_ films presented the highest ∆E value when compared to the NS precursor, S_5min_ and S_15min_, reaching 5.1 ± 0.7. Regarding blend films, the incorporation of S_15min_ or S_30min_ led to a significant ∆E increase in some blends, such as NS/S_15min_ (ratios 7:3 and 1:1) and NS/S_30min_ (ratio 3:7). The highest ∆E values were observed for NS/S_15min_ at 7:3 and 1:1 ratios, since the blend films presented higher b* values and less lightness, as described previously. No significant differences in ∆E values were observed following the addition of S_5min_ to the blend films. The total color difference (∆E) observed between the set of samples was discrete in comparison to other chitosan biobased films incorporating different additives, such as like tea polyphenols [19] or Aloe vera gel [20].

Opacity increased for all S films and most NS/S blend films, excluding NS/S_5min_ at a 1:1 ratio, NS/S_15min_ at a 3:7 ratio, and NS/S_30min_ at a 7:3 ratio, suggesting that former NS chitosan film was more transparent. The morphology of both the nanosized chitosan and the blend films, in general, were more homogeneous and compact than NS films which may explain, at least in part, the observed results. However, the opacity of the NS/S blend films did not significantly compromise transparency, still allowing for their use as food packaging.

### 2.5. Antimicrobial Activity of Chitosan Films

All films, composed by both the nonsonicated and sonicated chitosan (Figure 4A1,B1), as well as their blends (Figure 4A2,B2) displayed efficient antimicrobial activity against *Escherichia coli* and *Staphylococcus aureus*, Gram negative and Gram positive bacteria, respectively. The clear zones under the film cut indicate growth inhibition of the seeded bacteria. Medium- and low-sized chitosan and their nanosized counterparts have exhibited inherent antibacterial activity, with both MIC (minimum inhibitory concentration) and MBC (minimum bactericidal concentration) between 0.4–0.2 mg/mL against *Escherichia coli* DH5alpha [21]. Indeed, a discrete increased antibacterial activity has been displayed by the low molecular mass composites [22], which may make blend films advantageous when compared to unsonicated film precursors.

In addition to improvements in mechanical and barrier properties, chitosan blend films offer factual potential for application as bioactive packaging for perishable foodstuffs, improving food safety and food preservation. Particularly, the NS/S_15min_ films, at a 1:1 blend ratio, should be considered as an innovative flexible packaging solution, which may be applied as cup and tray lidding films and in the preservation of very perishable foodstuffs, such as sliced ham, fish and meat, and delicate fresh fruits, due to their particular physicochemical and mechanical properties and broad antimicrobial activity demonstrated herein. Alongside the aforementioned characteristics, nontoxicity allows for contact with ready-to-eat food, and ecofriendly film processing point towards a new generation of biobased chitosan displaying natural enhanced antimicrobial activity.

## 3. Materials and Methods

### 3.1. Materials and Solutions

A commercial ChitoClear chitosan supplied by Primex (Siglufjordur, Iceland) was used. This product is extracted from shrimp shells from the North Atlantic Ocean and further deacetylated to a degree of 90% (*DD* = 90%). The viscosity-average molecular mass (M_v_) is around 650 kDa at pH 6 [13]. This high-M chitosan is termed nonsonicated CHIT90 (NS).

Glacial acetic acid (Merck & Co., Inc., NJ, USA, purity: >99%), ultrapure sodium acetate trihydrate (Merck & Co., Inc.), and sodium chloride (NaCl > 99.9%) (Sigma-Aldrich Co., MO, USA) were all analytical grade and used without further purification.

### 3.2. Biopolymer Fragmentation and Blend Preparation

Chitosan degradation was carried out according to our previous study [14]. Briefly, using a SONIC ultrasonic probe, model 750 (Sonic & Material, Inc., CT, USA), equipped with a 1/2 microtip (constant duty cycle and 40% amplitude), 30 g of 3% (*w*/*w*) NS colloidal dispersion (in 0.1 M acetate buffer, pH 6) was placed on an ice bath and submitted to ultrasonic irradiation for defined periods of time (degradation time, t_S_ = 5, 15, or 30 min). Measurements were performed in triplicate.

Nonsonicated (NS) and sonicated (S) chitosan solutions have been previously fully characterized [13]. The results revealed a continuous decrease of the viscosity-average molecular mass (M_V_) from 660 kDa (NS) to 540 (S_5min_), 327 (S_15min_), and 291 kDa (S = 30 min). Morphological inspections indicate chitosan-like microfibers filled with nanoparticles (NPs), identified as NS chitosan (small- (10–30 nm) and medium-sized (150–300 nm) chitosan) [14]. Increasing sonication times led to alterations in NP shape and size. Five minutes of chitosan (CHIT90) ultrasonication formed ~15 nm nanosized particles and led to decreased decrease width of large fibers. Further decreases in particle size to ~12 nm were obtained by increasing chitosan (CHIT90) sonication to 30 min, leading to a more homogenous NP distribution.

Further support was obtained from the high polydispersity index derived from a dynamic light scattering analysis [14]. Three different intensity scattered peaks (microfibers, medium NPs, and small NPs) were observed for all samples, with a polydispersity index = 3. The contribution of the microfibers and NPs to light scattering depended on the sonication time. Under the assessed conditions, microfiber degradation mainly produced medium-sized NPs (220 ± 50 nm), while the smallest NPs (28 ± 10 nm) appear to be a minor sonication product.

For colloidal blend preparation, the appropriate weight of NS colloidal dispersion was added to aliquots of chitosan previously sonicated for different times (the sonicated component, or S). Different NS/S mixing ratios (7:3, 1:1, or 3:7) and different sonication times (t_S_ = 5, 15, or 30 min) were tested.

### 3.3. Biobased Film Fabrication

Chitosan-based films were prepared by the knife-coating method [13], where 30 g of 3% (*w*/*w*) colloidal film blends were spread over 0.3 m∙s^−1^ acrylic plate using a 1132N film applicator (TQCSheen Instruments, MI, USA), put in an environmental test chamber (T = 40 °C and relative humidity, RH = 53%) and dried for 18 h. Subsequently, films were peeled off from their supports, rinsed with distilled water, and stored for 18 h in the environmental test chamber.

Shorthand notations were adopted in the manuscript to designate the chitosan films prepared with nonsonicated chitosan (NS), sonicated chitosan (t_S_ = 5, 15, or 30 min), and their blends (NS/S) at 7:3, 1:1, or 3:7 ratios, respectively.

### 3.4. Film Characterization

#### 3.4.1. Thickness Measurements

Film thicknesses were determined at a 1 μm resolution using an ID-F150 Absolute Digimatic Indicator (Mitutoyo Co., Kanagawa, Japan). At least five measurements carried out at different positions were performed for each film.

#### 3.4.2. Mechanical Properties

Mechanical properties (tensile strength; Young’s modulus, and elongation at break) were determined using a TA.XT2 texture analyzer (Stable Micro Systems Ltd., Surrey, UK) equipped with tensile test attachments, according to the ASTM-D882 method [22].

The test specimens consisted of uniform width and length strips (25 mm × 100 mm) conditioned for 7 days until reaching constant weight in a climatic chamber (T = 25 °C and RH = 53%). Experiments were run immediately after removing the specimens from the climatic chamber, to minimize water adsorption/desorption. The distance between the grips was of 60 mm and the applied test speed was of 0.1 mm∙s^−1^, and force (N) and deformation (% strain) were recorded. Tests were performed on 5 replicates of each sample.

#### 3.4.3. Water Vapor Permeability

Permeability tests were performed according to the Standard Test Method for Water Vapor Transmission of Materials, ASTM2000 [23].

Transparent films, cut into 80 mm diameter discs, were equilibrated for 7 days in the environmental chamber (T = 25 °C, RH = 53%). Subsequently, they were tightly sealed in a permeation cell containing anhydrous calcium chloride (RH = 2%) and stored at room temperature in a container filled with distilled water (RH = 100%). Air convection was used to facilitate water diffusion. The cell was periodically weighed and water permeability, given as the water vapor transfer rate (*WVP*), in g∙m^−1^∙s^−1^∙Pa^−1^, was calculated according to Equation (1):*WVP* = (Δ*m*∙*x*)/(*A*∙Δ*t*∙Δ*P*)(1)
where Δ*m* is the weight gain (g), *x* is the film thickness (m), and *A* is the area (0.003 m^2^) exposed for a certain period of time Δ*t* (s) to a partial water vapor pressure Δ*p* (Pa).

#### 3.4.4. Water Sorption Isotherms

Water sorption isotherms were gravimetrically determined as described by Larotonda [24] and Sousa [25]. Briefly, films were cut into (25 × 25) mm^2^ pieces, dried under vacuum at 60 °C for 40 h and placed at 25 c into hermetic containers with equilibrium water activities (*a_W_*) ranging from 0.1 to 0.9. Samples were periodically weighed until constant weight (3–7 days). The water sorption isotherms were determined for each sample by the correlation of *a_W_* with moisture content, on a dry basis, *X*_e_. The Guggenheim–Anderson–de Boer (GAB) model was chosen to fit the water sorption data and provide additional thermodynamic information on the studied systems (Equation (2)).
*X_e_* = (*C*∙*k*∙*X*_0_∙*a_w_*)/([1 − *k*∙*a_w_*]∙[1 − *k*∙*a_w_* + *C*∙*k*∙*a_w_*])(2)

This is a typical model for fitting sigmoidal-shaped equilibrium moisture isotherms in foodstuffs [15,26], where, *X_0_* represents the monolayer moisture content, on a dry basis, and *C* is a constant, which depends on the difference between the sorption heats of the monomolecular and multimolecular layers. *k* is another constant reflecting the difference between the sorption heat of the multilayer and water vapor condensation heat. At least duplicate measurements were performed for each film formulation.

#### 3.4.5. Scanning Electron Microscopy

Scanning electron microscope images were acquired in the secondary electron mode on a Quanta 400 FEG microscope (FEI, OR, USA), located at CEMUP (Centro de Materiais da Universidade do Porto, Porto, Potugal). Samples were previously cryo-fractured using liquid N_2_ and mounted on aluminum stubs covered with double-coated carbon conductive adhesive tabs. Subsequently, the samples were coated with a thin film of Au/Pb and imaged in the high-vacuum/secondary electron imaging mode under an accelerating voltage of 10 kV, with working distances ranging between 10.4 and 13.8 mm.

#### 3.4.6. Color and Opacity

Color and opacity parameters were determined with a Minolta colorimeter CR300 series (Tokyo, Japan). CIE Lab color parameters, lightness (*L**) and chromaticity parameters *a** (red-green) and *b** (yellow-blue), were measured for each film. Film color was expressed as the difference of color (Δ*E**) according to Equation (3). $L_{S}^{*}$ (97.10), $a_{S}^{*}\;$(0.05), and $b_{S}^{*}$ (1.76) are CIE Lab standard values for the white standard were used as the film background [9].
(3)$$\Delta E^{*}=\sqrt{{\left(L^{*}-L_{s}^{*}\right)}^{2}+{\left(a^{*}-a_{s}^{*}\right)}^{2}+{\left(b^{*}-b_{s}^{*}\right)}^{2}}$$

Film opacity was determined according to the Minolta-lab method, in reflectance mode [27]. Opacity (*Y*) was calculated from the relationship between film opacities superimposed on the black standard (*Y*_black_) and on the white standard (*Y*_white_), according to Equation (4):(4)$$Y=\frac{Y_{\mathrm{black}}}{Y_{\mathrm{white}}}100$$

Five readings were performed for each film.

### 3.5. Antimicrobial Activity

Two common foodborne bacteria, *Escherichia coli* DH5alpha and *Staphylococcus aureus* ATCC14458 were grown in LB (Luria-Bertani) and BHI (Brain heart infusion) media, respectively, for 18 h at 250 rpm and 37 °C, at a concentration of 10^4^–10^5^ colony-forming units (CFU)/mL (estimated by the McFarland scale) and were used to seed LB and BHI plates.

Twenty grams of each nonsonicated chitosan (NS), sonicated chitosan (S), or blends (NS/S) solutions were then cast in 12 cm × 16 cm polyacrylic plates and dried at 25 °C for 24 h, for film formation.

The antimicrobial activity of each film was determined using the agar diffusion method. Chitosan films were cut into 10 × 10 mm^2^ square pieces and placed on Mueller Hinton agar plates previously seeded with 0.1 mL inoculum of indicator microorganisms ranging from 10^5^ to 10^6^ CFU/mL (estimated by the McFarland scale). The plates were then incubated at 37 °C for 24 h. The antimicrobial activity of each film blend was evaluated through the clear zones under film squares, indicating bacterial growth inhibition.

### 3.6. Statistical Analyses

Data were analyzed by the ANOVA test with Bonferroni post-test using the GraphPad Prism software package (San Diego, CA, USA), version 5.00 for Windows. To assess the means and differences between mean values, the significance level was defined at *p* < 0.05. The analyses were performed between films produced with NS chitosan, with sonicated chitosan (S) obtained after ultrasonication for different time periods (5, 15, or 30 min) and NS/S blends.

## 4. Conclusions

Chitosan and nanosized chitosan display inherent antibacterial properties and their film-forming ability make them an ideal choice for use as a biodegradable antimicrobial packaging material that can be used to improve the storability of perishable foods. The produced biobased films were chemically homogeneous and totally biodegradable, reinforcing their green design and ecofriendly nature.

The production of flexible films with adequate transparency and homogeneous morphology can be obtained by chitosan and nanosized chitosan blend. Fine-tuning the mechanical and water resistance properties of these films can be achieved by controlling sonication times and blend ratios between sonicated and nonsonicated components. Thus, these results provide strong support for the important role played by smart design on the production of a new generation of additive-free sustainable biobased films.

Besides tunable functional properties, all investigated films displayed efficient and broad antimicrobial activity against *Escherichia coli* and *Staphylococcus aureus*, Gram negative and Gram positive bacteria, respectively.

The production of high performance biobased films from fishery waste byproducts without the generation of toxic or secondary composites points to a possibility of replacing films made from petroleum-derived conventional polymers. In addition, the films applicator used herein to produce chitosan/nanosized chitosan film is commonly inserted in the plastic/film industry, facilitating film preparation on a larger scale. At the same time that several industrial activities besides food and pharmaceutical manufacturers increasingly demanding the development of nanomaterials, it is clear that ultrasonication will be price-industry compatible in the near future. It should also be noted that their antimicrobial activity is inherent to chitosan and chitosan nanoparticles, displaying an advantage that antimicrobial film activity does not depend on film manufacturing or the complexation to antimicrobial agents that certainly would increase the costs.

Chitosan and its nanoparticle films described herein unite desirable characteristics joining both physicochemical and microbiological performances, contributing to the improvement of research frameworks related to innovation and to the sustainable evolution of the food packaging industry.

## Figures and Tables

**Figure 1 molecules-24-00127-f001:**
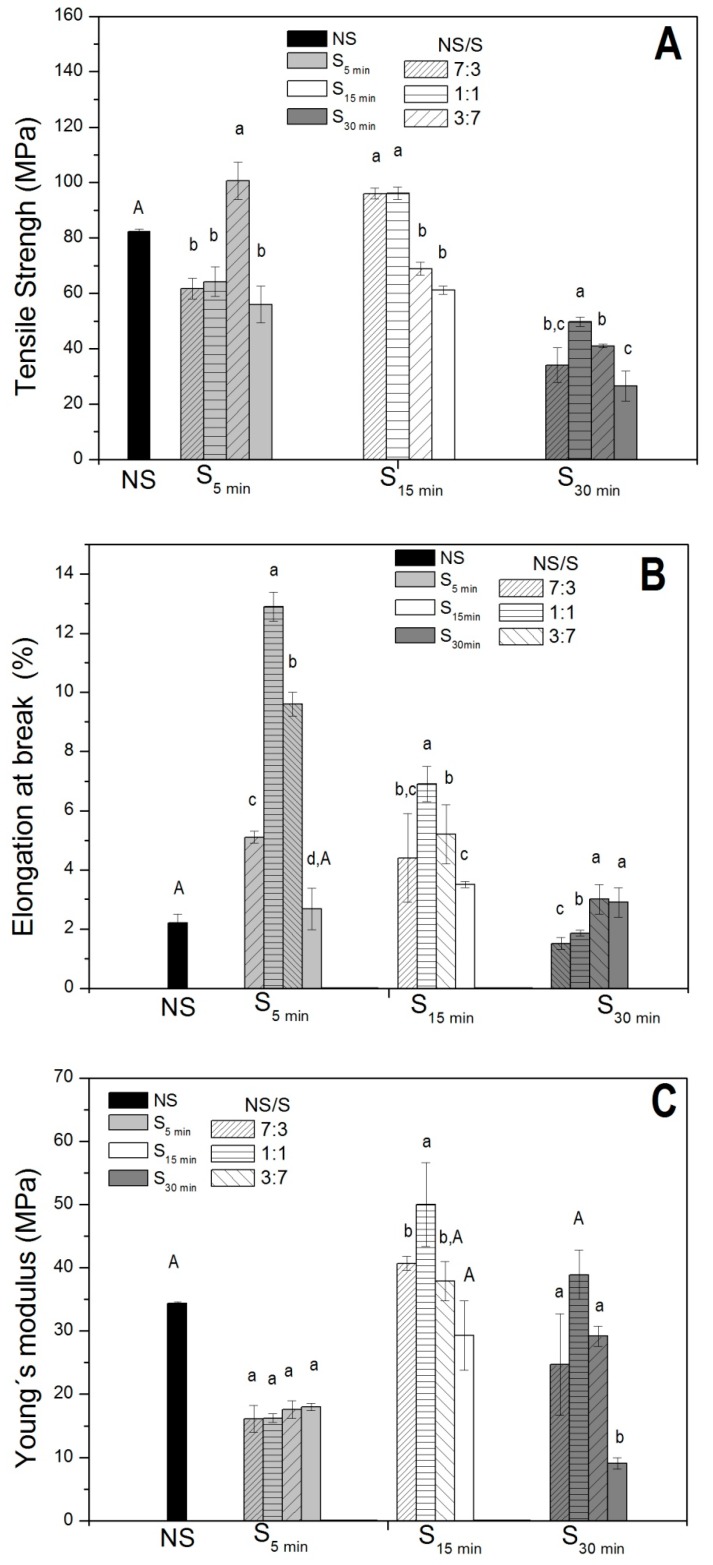
Mechanical properties of chitosan biobased films prepared from nonsonicated chitosan (NS) and nanosized chitosan sonicated at increasing times—5, 15, or 30 min (S_5min_, S_15min_ and S_30min_)—and NS/S blends. (**A**) Tensile strength, (**B**) elongation at break, and (**C**) Young’s modulus. NS/S 7:3, 1:1, and 3:7 ratios were evaluated for each film set. Same superscript letters within the same column indicate no differences at a significance level of *p* < 0.05 after the application of the Bonferroni test for each film set (S_5min_, S_10min_, and S_30min_).

**Figure 2 molecules-24-00127-f002:**
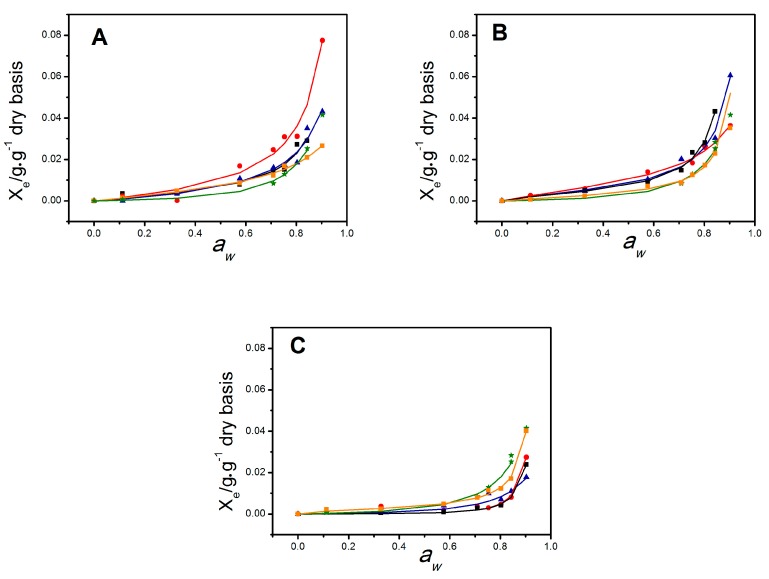
Experimental sorption data—equilibrium moisture content (X_e_) vs. water activity (a_w_). Dry basis symbols and the corresponding fitting (solid lines) using the GAB model for the NS, S, and NS/S biobase chitosan films. NS (green stars, all panels) and S (orange squares, all panels). The following NS/S ratios were studied for each system: 7:3 (black squares), 1:1 (red circles) and 3:7 (blue triangles) in all panels; nanosized chitosans were represented in panel **A** (S_5min_), panel **B** (S_15min_), and panel **C** (S_30min_). Films were cut into 25 × 25 mm^2^ pieces, dried under vacuum at 60 °C for 40 h and placed at 25 °C into hermetic containers with equilibrium water activities (*a_W_*) ranging from 0.1 to 0.9.

**Figure 3 molecules-24-00127-f003:**
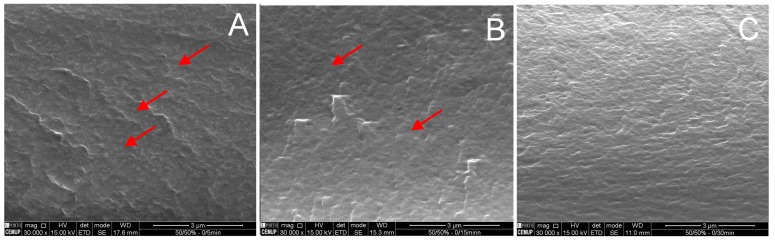
Representative scanning electron microscope images of the cross-sections of cryo-fractured NS/S blend films at a 1:1 ratio, 30,000× magnification. Panel **A**—NS/S_5min_, Panel **B**—NS/S_15min_, and Panel **C**—NS/S_30min_. Red arrows indicate nanosized chitosan molecules.

**Figure 4 molecules-24-00127-f004:**
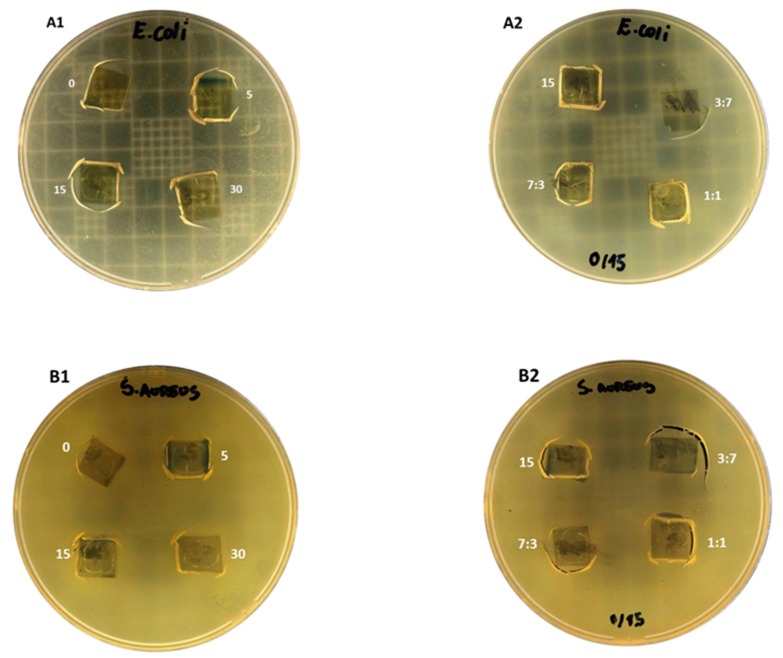
Inherent antimicrobial activity of biobased films. Chitosan and nanosize chitosan (S_5min_, S_15min_, and S_30min_) films were tested against *Escherichia coli* (**A1** and **A2**) and *Staphylococcus aureus* (**B1** and **B2**) bacteria. In Panel 1, **A1** and **B1** designate films composed of NS (0) or S obtained after 5, 15 and 30 min sonication times. In Panel 2, **A2** and **B2**, numbers designate films comprising S_15min_ sonicated chitosan (15) and blend films comprising NS/S_15min_ at 3:7, 1:1, and 7:3 ratios. Square film cuts were placed on Mueller Hinton agar plates previously seeded with 10^5^–10^6^ CFU/mL indicator microorganisms. Inhibition of bacterial growth under the film squares were evaluated after incubation at 37 °C for 24 h.

**Table 1 molecules-24-00127-t001:** Film thickness (*d*), water vapor permeability (WVP) and GAB parameters obtained from fitting of the data displayed in Figure 2 to Equation (2) for biobased films prepared from nonsonicated chitosan (NS), sonicated chitosan (S), and NS/S blends.

Composites	Blend Ratio	*d*/mm	WVP∙10^−11^/g.m^−1^s^−1^Pa^−1^	GAB Parameters			
				C	k	X_0_	R^2^
NS	-	0.028 ± 0.00 ^A^	14.4 ± 0.01 ^A^	0.017	0.9	0.12	0.991
NS/S_5min_	7:3	0.023 ± 0.002 ^a^	7.9 ± 1.1 ^a^	2.84	1.0	0.005	0.981
	1:1	0.025 ± 0.003 ^a^	9.5 ± 0.4 ^a^	1.62	1.0	0.01	0.991
	3:7	0.020 ± 0.001 ^a^	5.9 ± 0.7 ^b^	1.08	1.0	0.01	0.982
S_5min_	-	0.008 ± 0.001 ^b^	10.8 ± 0.9 ^a^	2.82	0.9	0.01	0.997
NS/S_15min_	7:3	0.020 ± 0.001 ^a^	13.2 ± 3.2 ^A,a^	5.02	1.07	0.01	0.995
	1:1	0.017 ± 0.002 ^a^	12.1 ± 0.13 ^a^	3.46	0.88	0.01	0.995
	3:7	0.022 ± 0.003 ^a^	9.1 ± 0.7 ^a^	2.24	0.95	0.10	0.999
S_15min_	-	0.008 ± 0.0001 ^b^	7.8 ± 0.0 ^a^	3.09	1.05	0.003	0.997
NS/S_30min_	7:3	0.013 ± 0.002 ^a^	8. 7 ± 0.6 ^a^	0.002	1.0	0.1	0.993
	1:1	0.018 ± 0.001 ^a^	6. 3 ± 2.4 ^a^	0.002	1.0	0.1	0.984
	3:7	0.011 ± 0.001 ^b^	3.9 ± 0.04 ^b^	0.013	0.9	0.1	0.973
S_30min_	-	0.010 ± 0.002 ^b^	7.4 ± 0.02 ^a^	11.6	1.0	0.002	0.998

Values are expressed as means ± standard deviations. Nonsonicated chitosan—NS, sonicated chitosan for 5, 15, or 30 min—S_5min_, S_15min_, or S_30min_, respectively. Means alongside an uppercase letter (A) indicate no difference between NS and S or NS/S at a significance level of *p* < 0.05 after the application of the Bonferroni test. Same superscript letters within the same column indicate no differences at a significance level of *p* < 0.05 after the application of the Bonferroni test for each film set (S_5min_, S_10min_, and S_30min_).

**Table 2 molecules-24-00127-t002:** Effect of nanosized chitosan (S) incorporation to the chitosan solution (NS) at different blend ratios on blend films color characteristics—a*, b*, L*, ΔE*, and opacity.

	Blend Ratio	L*	a*	b*	ΔE*	Opacity
NS	-	95.67 ± 0.2 ^A^	−0.302 ± 0.04 ^A^	3.26 ± 0.2 ^A^	2.1 ± 0.3 ^A^	10.4 ± 0.2 ^A^
NS/S_5min_	7:3	97.22 ± 0.2 ^a^	−0.21 ± 0.1 ^b^	3.05 ± 0.8 ^A,c^	1.3 ± 0.8 ^A,b^	-
	1:1	95.38 ± 0.1 ^A,c^	−0.27 ± 0.06 ^b^	3.4 ± 0.2 ^b^	2.4 ± 0.2 ^A,a^	12.2 ± 1.9 ^A,a^
	3:7	96.88 ± 0.8 ^b^	−0.33 ± 0.3 ^A,c^	3.9 ± 1.7 ^a^	2.3 ± 1.8 ^A,a^	-
S_5min_	-	96.17 ± 0.5 ^A,b^	0.002 ± 0.008 ^a^	2.6 ± 0.1 ^d^	1.3 ± 0.4 ^b^	12.3 ± 0.8 ^a^
NS/S_15min_	7:3	91.30 ± 0.8 ^c^	−0.53 ± 0.06 ^c^	5.52 ± 0.2 ^a^	7.0 ± 0.7 ^a^	13.4 ± 0.6 ^a^
	1:1	92.65 ± 0.8 ^b^	−0.52 ± 0.07 ^c^	5.05 ± 0.2 ^b^	5.6 ± 0.7 ^b^	13.2 ± 0.4 ^a^
	3:7	95.24 ± 1.0 ^A,a^	−0.26 ± 0.04 ^A,b^	3.72 ± 0.2 ^c^	2.8 ± 0.8 ^A,c^	10.4 ± 1.5 ^A,b^
S_15min_	-	96.25 ± 0.4 ^A,a^	0.07 ± 0.04 ^a^	2.57 ± 0.3 ^d^	1.8 ± 0.5 ^A,d^	12.3 ± 0.7 ^a^
NS/S_30min_	7:3	95.01 ± 0.6 ^A,a^	−0.11 ± 0.1 ^a^	3.39 ± 0.4 ^A,c^	2.6 ± 0.8 ^A,c^	10.7 ± 0.9 ^A,b^
	1:1	95.86 ± 0.7 ^A,a^	−0.18 ± 0.08 ^a^	3.42 ± 0.2 ^A,c^	2.1 ± 0.5 ^A,c^	12.2 ± 1.1 ^a^
	3:7	94.63 ± 0.7 ^b^	−0.32 ± 0.04 ^A,b^	3.90 ± 0.3 ^b^	3.3 ± 0.7 ^b^	12.0 ± 1.1 ^a^
S_30min_	-	93.34 ± 0.6 ^b^	−0.56 ± 0.06 ^c^	5.2 ± 0.3 ^a^	5.1 ± 0.7 ^a^	12.6 ± 0.6 ^a^

Values are expressed as means ± standard deviations. A* (red-green) and b* (yellow-blue). The values with an uppercase letter (A) indicate no a significant difference when compared to NS. Different letter (a, b and c) superscripts within column indicate differences at a significance level of *p* < 0.05.

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
