# Peer review of "Edible Chitosan Films and Their Nanosized Counterparts Exhibit Antimicrobial Activity and Enhanced Mechanical and Barrier Properties"

_molecules, 2018, doi:10.3390/molecules24010127_

Round 1
Reviewer 1 Report
The fabrication of transparent chitosan bio-based films for food packaging is a smart application of this material
Author describe an innovative procedure to manufacture edible food package films with fine-tuned mechanical properties, nice optical appearance (transparences) without the inclusion of any plasticizers that could introduce toxic or non-biodegradable components to the chitosan films. In addition they demonstrate a general antimicrobial properties of the films developed against E. coli and S. aureus.
The paper is clear and well written. Materials are widely and soundly characterized. Conclusions are supported by the results
Author Response
We believe we have fully addressed the editor(s)/reviewer(s) concerns. The text was modified as suggested and the modifications were highlighted in yellow.
We have provided a point-by-point response to the all items in the comments for authors that will help the editor(s)reviewers(s) gauge the changes made in the revised manuscript.
We thank the editor(s)/reviewer(s) for their insight and thoughtful critique of our manuscript, which has increased the overall quality of the manuscript.
Reviewer 2 Report
The topic meet the journal, by the way the quite high impact of the journal request for very sound and original papers. In the present work is not easy to understand the innovation versus previously published papers both of the authors than of other research groups. Chitosan filming ability and anti-microbial properties are very well know, thus the originality and value of the present work needs to be better focused and outlined.
Better and clearly stress the innovation and new funding respect to previous paper as in ref 13, and 14. Or to very similar paper in the literature (ex: Physical and mechanical properties of chitosan films as affected by drying methods and addition of antimicrobial agent, WasinaThakhiewa, SakamonDevahastin, SomchartSoponronnarita,https://doi.org/10.1016/j.jfoodeng.2013.05.020).
The authors refer the present material to possible practical applications.
Have you done an estimation of the cost of your films, and the possibility to industrialize their production? Have you considered applying the material as a coating on a polymeric substrate?
Bio based films fabrication. Since you propose applications in packaging it is not clear how can this fabrication procedure be converted in something suitable for a large scale production?
The values of tensile tests reported are very high, may you report an example of a strain to stress curve?
Minor spelling errors
Line 139 “an structure” change in “a structure”
Phrase 153 is confused. Re phrase and explain better what do you mean? Which films application do you target? Films for agriculture, are not produce by casting. What do you mean for “biotechnology inputs”?
Did you perform any tests on bio-compatibility?
Line 406: do you mean: test was performed on 5 replicates for each sample?
Author Response
We believe we have fully addressed the editor(s)/reviewer(s) concerns . The text was modified as suggested and the modifications were highlighted in yellow. We have provided a point-by-point response to the all items in the comments for authors that will help the editor(s)reviewers(s) gauge the changes made in the revised manuscript. We thank the editor(s)/reviewer(s) for their insight and thoughtful critique of our manuscript, which has increased the overall quality of the manuscript.
We believe that we have fully addressed and understood all of Reviewer 2 concerns and comments.
A supplementary figure was included – Figure S1 – as an example of a stress curve (reviewer 2).
Figure 2 was altered to a colored version in order to better display the data, as recommended (reviewer 3).
Figure 3 was reduced to 75% of its original size, as recommended (reviewer 3).
After modifications, the manuscript was revised by a specialized editing company in order to improve English grammar and syntax.
The modifications have increased the overall impact of the manuscript. We would like to thank the editor/reviewer for his/her insights and thoughtful critique of our manuscript.
We hope that in its revised form our manuscript will be found suitable for publication in this reputable journal.

Reviewer 3 Report
Comments:
-l.70-79: usually “The aim of...” is a legible
-l.94: “sonicated precursors” – better; “sonicated precursors (S)”
-l.160-180: discussion on variability of WVP – see Tab. 1 – significantly higher WVP was obtained only for NS/S15 min and blend ratio 7:3. The same influence for C as well as for all colour properties (Tab. 2). It should be clearly explained
-l. 174:”…<25 min…” why 25 ?
-l.181: “Temperature ..” – why ?
-l. 184: “dramatically” – better not so strong
-l. 195: literature should be cited
-l. 205 – 213: Fig.2 – should be much more a legible ( especially symbols), e.g. for NS open triangles and also for NS/S - 3:7
-l. 247 – 255: images could be much smaller, e.g. 25 % of each image
-l.259 – 263: could be omitted
-l. 402 and l. 414: RH=53 % was chosen
-l. 403: (3-7 days) – short time to obtain equilibrium – should be explained
-l. 519: “Altogether and from a practical point of view” – could be omitted
-l. 524 – 527: the sentence could be omitted
-l. 503 – 530: all abbreviations, like e.g. CS, NS and S could be cut off from the conclusions
Author Response
We believe we have fully addressed the editor(s)/reviewer(s) concerns . The text was modified as suggested and the modifications were highlighted in yellow. We have provided a point-by-point response to the all items in the comments for authors that will help the editor(s)reviewers(s) gauge the changes made in the revised manuscript. We thank the editor(s)/reviewer(s) for their insight and thoughtful critique of our manuscript, which has increased the overall quality of the manuscript.
We believe that we have fully addressed and understood all of Reviewer 3 concerns and comments.
A supplementary figure was included – Figure S1 – as an example of a stress curve (reviewer 3).
Figure 2 was altered to a colored version in order to better display the data, as recommended (reviewer 1).
Figure 3 was reduced to 75% of its original size, as recommended (reviewer 1).
After modifications, the manuscript was revised by a specialized editing company in order to improve English grammar and syntax.
The modifications have increased the overall impact of the manuscript. We would like to thank the editor/reviewer for his/her insights and thoughtful critique of our manuscript.
We hope that in its revised form our manuscript will be found suitable for publication in this reputable journal.
